# OVSeg3R: Learn Open-vocabulary Instance Segmentation from 2D via 3D Reconstruction

**Hongyang Li** *
South China University of Technology
International Digital Economy Academy (IDEA)

**Jinyuan Qu** *
Tsinghua University
International Digital Economy Academy (IDEA)

**Lei Zhang** †
South China University of Technology
International Digital Economy Academy (IDEA)

## Abstract

In this paper, we propose a training scheme called OVSeg3R to learn open-vocabulary 3D instance segmentation from well-studied 2D perception models with the aid of 3D reconstruction. OVSeg3R directly adopts reconstructed scenes from 2D videos as input, avoiding costly manual adjustment while aligning input with real-world applications. By exploiting the 2D to 3D correspondences provided by 3D reconstruction models, OVSeg3R projects each view's 2D instance mask predictions, obtained from an open-vocabulary 2D model, onto 3D to generate annotations for the view's corresponding sub-scene. To avoid incorrectly introduced false positives as supervision due to partial annotations from 2D to 3D, we propose a View-wise Instance Partition algorithm, which partitions predictions to their respective views for supervision, stabilizing the training process. Furthermore, since 3D reconstruction models tend to over-smooth geometric details, clustering reconstructed points into representative super-points based solely on geometry, as commonly done in mainstream 3D segmentation methods, may overlook geometrically non-salient objects. We therefore introduce 2D Instance Boundary-aware Superpoint, which leverages 2D masks to constrain the superpoint clustering, preventing superpoints from violating instance boundaries. With these designs, OVSeg3R not only extends a state-of-the-art closed-vocabulary 3D instance segmentation model to open-vocabulary, but also substantially narrows the performance gap between tail and head classes, ultimately leading to an overall improvement of +2.3 mAP on the ScanNet200 benchmark. Furthermore, under the standard open-vocabulary setting, OVSeg3R surpasses previous methods by about +7.1 mAP on the novel classes, further validating its effectiveness.

## 1 Introduction

Recent advances in 3D reconstruction (Murai et al., 2025; Wang et al., 2024; 2025) have made scene geometry capturing accessible. Yet downstream tasks such as manipulation (Liu et al., 2024c; Black et al., 2024), navigation (Song et al., 2025), and augmented reality (AR) require recognizing objects with instance-level identities and locations. Such demand has driven a growing interest in 3D instance segmentation and its open-vocabulary generalization.

Despite major progress in 2D segmentation (Li et al., 2023), where open-vocabulary capabilities already meet most downstream demands (Ren et al., 2024), 3D instance segmentation capabilities remain limited. This limitation persists even though the cost of acquiring 3D scenes has been greatly reduced by the remarkable progress of 3D reconstruction techniques, as the cost of acquiring 3D annotations is still expensive

---

*Equal contribution. This work was done while Hongyang Li and Jinyuan Qu were interns at IDEA.
†Corresponding author.

Figure 1: (a) Traditional training scheme relies on costly manual efforts and non-routine sensors, such as depth cameras, to construct training data and annotations. (b) OVSeg3R leverages modern 3D reconstruction models and well-studied 2D perception models to construct training data and annotations. To further alleviate the issue of partial supervision and over-smoothness to improve training stability, we use the 2D-3D correspondences from 3D reconstruction models to partition scene-level predictions to assist supervision, and leverage 2D instance masks to constrain the superpoints, assisting segmentation.

Therefore, how to leverage diverse 3D scenes provided by 3D reconstruction models and robust 2D masks provided by well-studied 2D perception models to enhance 3D instance segmentation has become an important research topic. Some approaches (Takmaz et al., 2023) train 3D segmentation models solely to produce class-agnostic 3D masks, which are then projected onto 2D to retrieve category information from 2D foundation models (Radford et al., 2021). While this strategy can effectively exploit the strong classification ability of 2D models, it remains limited by the scarcity of 3D annotations, making it difficult to generate reliable 3D masks for unseen objects and often resulting in missed detections. Other methods (Yang et al., 2023) project each view's 2D segmentation results (Kirillov et al., 2023) into 3D space using the 2D pixel to 3D point correspondences provided by a 3D reconstruction model, and merge the projected masks that belong to the same instance through heuristic strategies. Although this line of work benefits from the classification and segmentation strengths of 2D models, the hand-crafted merging process is error-prone, rendering these methods fragile and performance-constrained. While these methods demonstrate open-vocabulary potential, they over-rely on 2D outputs, leaving native 3D perception ability underdeveloped, which is crucial for advancing 3D understanding (Peng et al., 2023; Ding et al., 2023). Although some works try to distill from 2D models, they need to train 3D Gaussian (Kerbl et al., 2023) to create 2D and 3D correspondences, which is redundant since reconstruction inherently provides this. Moreover, these methods either require per-scene optimization (Lyu et al., 2024; Ye et al., 2024) or high-quality point cloud for Gaussian initialization (Cao et al., 2025), limiting their practicality.

To address the 3D annotation challenges, in this work, we propose a novel training scheme, called OVSeg3R. As shown in Fig 1, instead of relying on manually adjusted 3D scenes from non-routine sensors, OVSeg3R directly leverages a modern 3D reconstruction model such as (Murai et al., 2025) or (Wang et al., 2025) to provide point cloud inputs. This not only substantially reduces the cost of acquiring 3D scenes, but also naturally introduces noise to the inputs, aligning with application scenarios, where user-provided inputs are typically videos or low-quality reconstructions captured by handheld sensors. For annotations, instead of manually annotating 3D masks for the reconstructed scene, the inherent 2D pixel-to-3D point correspondence provided by 3D reconstruction enables us to lift the 2D masks generated by an open-vocabulary 2D segmentation model (Ren et al., 2024) from each view into 3D space to obtain view-level annotations. However, although 2D masks from different views may correspond to the same object, they are estimated independently and lack cross-view associations (Yang et al., 2023). Directly concatenating them to a scene-level annotation would introduce many duplicate annotations. Conversely, since each view covers only part of the reconstructed point cloud, supervising scene-level predictions with each view's partial annotations alone would incorrectly penalize predictions outside the view. To mitigate this issue, we propose a View-wise Instance Partition (VIP) algorithm. According to the visibility of each scene-level mask prediction's corresponding object query in different views, VIP assigns the mask predictions to their corresponding views. Then, for each mask prediction, VIP further truncates it to retain only the region visible within its belonging view. The resulting view-level predictions enable supervision with view-specific annotations, which eliminates annotation duplication and avoids incorrect penalization, thus improving training stability.

Moreover, to improve efficiency, mainstream 3D instance segmentation models (Kolodiazhnyi et al., 2024b; Qu et al., 2025) normally leverage superpoints (Landrieu & Simonovsky, 2018), first over-segmenting the input point cloud into superpoints and then performing instance segmentation at the superpoint level. However, in OVSeg3R, 3D reconstruction results are often over-smoothed (Yang

et al., 2024), leading to the loss of geometric details. As a result, constructing superpoints purely based on geometric continuity, as in the previous method, may cause objects that are not geometrically salient, such as paintings, being included in a superpoint covering large planar regions, such as walls, as shown in Fig 3 (b). This will inevitably lead to inaccurate segmentation. To mitigate this issue, we propose the 2D Instance Boundary-aware Superpoint (IBSp), IBSp incorporates 2D instance masks into the construction of superpoints, avoids the erroneous clustering of points from different instances into the same superpoint, further stabilizes the training process.

With these designs, OVSeg3R enables the learning of open-vocabulary 3D instance segmentation from 2D models directly, without requiring any additional model capacity during training or fully relying on 2D model outputs during inference. The experimental results show that OVSeg3R can not only extend a closed-vocabulary model to open-vocabulary, but also, thanks to the strong category generalization it provides, significantly reduce the performance gap between tail and head classes. Consequently, it achieves an overall improvement of about +2.3 mAP on ScanNet200, surpassing all previous methods. Moreover, on the standard open-vocabulary setting, OVSeg3R achieves a significant improvement of +7.7 mAP on the novel classes, further validating the effectiveness.

In summary, our contributions are threefold:

- The main contribution of this work lies in the proposed training scheme OVSeg3R, which makes full use of the well-studied 3D reconstruction and 2D segmentation models to enable the training of end-to-end open-vocabulary 3D instance segmentation.
- To guarantee the training stability, we propose the View-wise Instance Partition (VIP) algorithm to prevent incorrect false positives, and the 2D Instance Boundary-aware SuperPoint (IBSp) to prevent the points of different objects from being clustered into the same superpoint.
- OVSeg3R extends a closed-vocabulary 3D instance segmentation model to open-vocabulary, achieving +2.3 mAP on ScanNet200 and +7.7 mAP on novel classes in the standard open-vocabulary setting, verifying its effectiveness.

## 2 RELATED WORKS

**Closed-vocabulary 3D Instance Segmentation**. Early 3D instance segmentation methods fall into two lines: proposal-based methods (Yang et al., 2019; Hou et al., 2019; Yi et al., 2019; Engelmann et al., 2020; Kolodiazhnyi et al., 2024a), which first detect objects and then refine 3D masks within the predicted bounding boxes, and grouping-based methods (Liang et al., 2021; Chen et al., 2021; Vu et al., 2022; Jiang et al., 2020b; Wang et al., 2019; Jiang et al., 2020a; Zhang & Wonka, 2021), which aggregate points via voting in feature or geometric space. Following the success of Detection Transformers (DETR) (Carion et al., 2020; Liu et al., 2021; Li et al., 2022; Zhang et al., 2022) in 2D, recent works (Sun et al., 2023; Schult et al., 2023; Lai et al., 2023; Kolodiazhnyi et al., 2024b; Jain et al., 2024) adopt DETR-like architectures for 3D instance detection and segmentation. Notably, SegDINO3D (Qu et al., 2025) proposes to leverage high-quality image- and object-level features from well-studied 2D models (Ren et al., 2024; Liu et al., 2024b) to support data-hungry 3D models, achieving substantial performance gains.

**Open-vocabulary 3D Instance Segmentation**. Motivated by the rapid progress in 2D open-vocabulary perception, most methods attempt to obtain open-vocabulary 3D perception outputs by relying on the outputs of 2D models. Generally, the mainstream methods can be categorized into two types. OpenMASK3D (Takmaz et al., 2023) first proposes to generate class-agnostic 3D segmentation results using 3D models (Schult et al., 2023). Then project each 3D result to 2D to obtain the corresponding object's category using 2D foundation models (Radford et al., 2021), thereby constructing an open-vocabulary 3D segmentor. While this approach achieves remarkable results and has inspired a series of subsequent works (Nguyen et al., 2024; Boudjoghra et al., 2025; Nguyen et al., 2025), it only provides classification capability for novel categories. Limited by the scarcity of 3D segmentation data, such methods often fail to segment objects that are unseen during training. SAM3D (Yang et al., 2023) is the first to lift 2D segmentation results from SAM for each frame to 3D and then merge those belonging to the same instance in 3D using a proposed heuristic algorithm. Motivated by the promising performance, many subsequent works (Yin et al., 2024; Lu et al., 2023; Zhao et al., 2025; Xu et al., 2025) have focused on improving this merging process. While these algorithms can reuse both the detection and classification capabilities of well-studied 2D models, the heuristics are fundamentally rule-based, lack generalization, and fail to handle numerous corner cases encountered in practical scenarios.

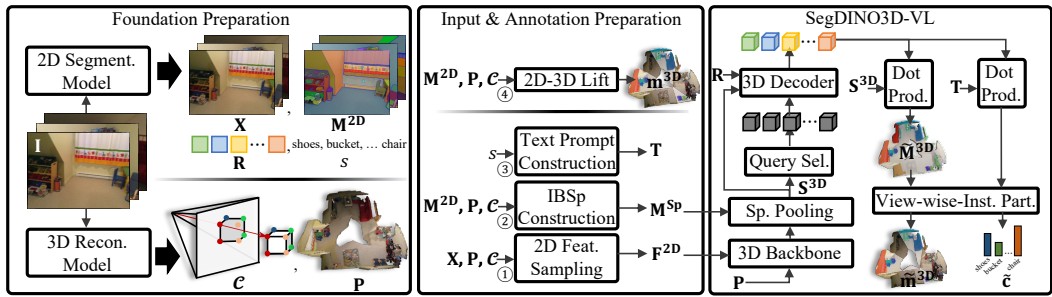

Figure 2: Training SegDINO3D-VL with OVSeg3R. Given an input video, we first apply 3D reconstruction and the 2D instance segmentation to prepare the foundation data. The prepared foundation will further be combined to construct the input and also the view-wise supervision for the 3D instance segmentator SegDINO3D-VL. The reconstructed scene is then fed into SegDINO3D-VL to produce the scene-level instance segmentation results, which are further partitioned to each view by the view-wise instance partition module for stable supervision. Please refer to Sec. 3 for the detailed definitions of mathematical symbols.

## 3 METHOD

To validate the effectiveness of the proposed training scheme, we adopt SegDINO3D (Qu et al., 2025), a recent state-of-the-art method, as our baseline. To further satisfy the requirement of open-vocabulary, we extend the classification part of SegDINO3D to the similarity calculation between object features and the text features, yielding SegDINO3D-VL. In this section, we will first describe the training of SegDINO3D-VL with OVSeg3R, including the preparation of data, construction of view-wise annotation, and obtaining predictions from SegDINO3D-VL. After that, we will describe in detail our designs for stable training.

### 3.1 TRAINING WITH OVSEG3R

#### 3.1.1 FOUNDATION PREPARATION

Given a video with $V$ views $\mathbf{I} \in \mathbb{R}^{V \times H \times W \times 3}$, where $H$ and $W$ are height and width, OVSeg3R feeds it to a 3D reconstructor and a 2D instance segmentator, instantiated as MASt3R-SLAM (Murai et al., 2025) and DINO-X (Ren et al., 2024) respectively by default, to prepare the foundation data.

**3D Foundation**. Given a video, the 3D reconstructor produces a point cloud of the corresponding scene, denoted as $\mathbf{P} \in \mathbb{R}^{N \times 3}$ with $N$ points, along with a 2D pixel to 3D point correspondence record $\mathcal{C}$. Specifically, $\mathcal{C}$ records bidirectional mappings between a 3D point's index and the point's corresponding view index and pixel coordinates, from which the 3D point is reconstructed:

$$\mathcal{C} : i \leftrightarrow (v, x, y), \quad i \in \{0, \ldots, N-1\}; v \in \{0, \ldots, V-1\}, \ (x, y) \in [0, 1]^2, \tag{1}$$

where $i$ and $v$ are the point index and the view index, and $(x, y)$ are the normalized pixel coordinates.

**2D Foundation**. For the 2D segmentators, we not only require the detected objects' class names $s$ (concatenated into a single string with dots "."), as well as the decoded per-view instance masks $\mathbf{M^{2D}} \in \mathbb{Z}^{V \times H \times W}$, where each pixel is assigned a 2D instance index, but following SegDINO3D, we need to prepare its intermediate feature representations. Specifically, we prepare the encoded image-level 2D features $\mathbf{X} \in \mathbb{R}^{V \times h \times w \times C}$ and the decoded object-level 2D features $\mathbf{R} \in \mathbb{R}^{O \times C}$ of 2D segmentators for SegDINO3D-VL, to enhance its 3D representation, where $h$ and $w$ are the size of the feature maps, $C$ is the feature dimension, and $O$ is the total number of detected 2D objects across $V$ views.

#### 3.1.2 INPUT AND VIEW-WISE ANNOTATION PREPARATION

After preparing the foundation date, we construct 3D instance segmentator's input and annotation.

**2D Feature for Each 3D Point**. Following SegDINO3D (Qu et al., 2025), we need to sample 2D image-level feature $\mathbf{F^{2D}} \in \mathbb{R}^{N \times C}$ for each 3D point. However, since the 2D-3D correspondences $\mathcal{C}$ are already available, re-projecting every 3D point to all views to identify a representative view, as required in SegDINO3D, becomes unnecessary. As described in ① of Fig 2, based on each 3D point's corresponding view index and the sampling location provided by $\mathcal{C}$, we sample 2D features

Figure 3: (a) Visualization of the 2D instance masks obtained from the well-studied 2D segmentators and their corresponding lifted view-wise 3D instance segmentation annotations. (b) Visual comparison between the superpoint built solely upon geometric continuity (geo.-only) and the proposed IBSp. Due to the over-smoothed nature of reconstructed results, geo.-only superpoints tend to cluster geometrically less salient objects (picture, power outlet, carpet) into their background, preventing them from being segmented out. By incorporating 2D instance boundaries, at least one superpoint is preserved for such objects (highlighted by the red arrows), mitigating the issue.

from 2D image feature maps $\mathbf{X}$ for the point through bilinear interpolation. For the sampling of $\mathbf{F}_i^{\mathbf{2D}} \in \mathbb{R}^C$ for the $i$-th 3D point, the process can be formulated as

$$v_i, x_i, y_i \Leftarrow \overrightarrow{\mathcal{C}}(i), \mathbf{F}_i^{\mathbf{2D}} \Leftarrow \mathrm{Bili}\left(\mathbf{X}_{v_i}, (x_i, y_i)\right), \tag{2}$$

where $\overrightarrow{\mathcal{C}}$ is the function that, given the 3D point index, outputs the triplet of the point's corresponding view index and the $x$ $y$ coordinates on the view according to $\mathcal{C}$.

**Superpoint Construction**. We use a superpoint mask $\mathbf{M}^{\mathbf{sp}} \in \mathbb{B}^{n \times N}$ to represent the clustering of $N$ points into $n$ superpoints, where each entry is a Boolean value, indicating whether a point belongs to a given superpoint (② in Fig.2). To obtain a better clustering, we propose the 2D Instance Boundary-aware Superpoint (IBSp), which considers not only the geometric continuity of the reconstructed points but also the 2D instance boundaries, preventing points from different objects from being clustered into the same superpoint. More details are provided in Sec. 3.3 and Fig. 3 (b).

**Text Prompt Feature**. Following previous methods (Liu et al., 2024b), we need to prepare the text features for each class name in $s$ for instance classification (③ in Fig 2). To enable contrastive learning and batch-friendly training, we randomly sample $T - |s|$ additional class names not present in $s$ as negative classes, padding $s$ to a fixed size $T$. The padded $s$ is then concatenated into a string (e.g., 'book . sofa .') and encoded with a text encoder, to produce text features $\mathbf{T} \in \mathbb{R}^{T \times C}$.

**View-wise Annotation**. As described in Fig 3 (a), we assign each pixel of the 2D instance mask $\mathbf{M}^{\mathbf{2D}}$ to its corresponding 3D point according to the 2D-3D correspondence $\mathcal{C}$, to obtain the 3D instance masks $\mathbf{m}^{\mathbf{3D}} \in \mathbb{Z}^{V \times HW}$. However, for the $v$-th view, the resulting view-wise 3D instance masks $\mathbf{m}_v^{\mathbf{3D}} \in \mathbb{Z}^{HW}$ only record the annotation of a subset of the scene. Thus, they can not be utilized to supervise the 3D segmentator's scene-level predictions directly; otherwise, the predictions that are outside the $v$-th view will be incorrectly supervised as false positives, leading to unstable training. To mitigate this issue, we propose the View-wise Instance Partition in Sec 3.2.

### 3.1.3 Obtain Predictions from SegDINO3D-VL

As in SegDINO3D, we send the 3D point cloud $\mathbf{P}$ and its corresponding 2D features $\mathbf{F}^{\mathbf{2D}}$ into the 3D backbone to extract the 3D point-level features $\mathbf{F}^{\mathbf{3D}} \in \mathbb{R}^{N \times C}$. The point-level features are further pooled into superpoint-level features $\mathbf{S}^{\mathbf{3D}} \in \mathbb{R}^{n \times C}$ by

$$\mathbf{S}^{\mathbf{3D}} \Leftarrow \mathbf{M}^{\mathbf{sp}} \mathbf{F}^{\mathbf{3D}} / \mathrm{Sum}\left(\mathbf{M}^{\mathbf{sp}}, 1\right), \tag{3}$$

where $\mathrm{Sum}(\cdot, 1)$ is the summation operation, along the 1-th dimension. After that, $q$ superpoints are selected as the initial 3D object queries $\mathbf{Q} \Leftarrow \mathbf{S}^{\mathbf{3D}}[\mathbf{q}], \mathbf{Q} \in \mathbb{R}^{q \times C}$, where $\mathbf{q} \in \mathbb{Z}^q$ is the selection indices, and sent to the multi-layer transformer decoder to refine their feature representation. In each layer of the mask decoder, the 3D object queries cross-attend to the superpoint features $\mathbf{S}^{\mathbf{3D}}$ and the 2D object features $\mathbf{R}$ sequentially, followed by a self-attention among the 3D object queries and a feed-forward MLPs to update object queries' content features. Finally, the scene-level 3D instance

masks $\widetilde{\mathbf{M}}^{\mathbf{3D}} \in \mathbb{B}^{q \times n}$ are decoded by thresholding the similarity map between the object queries and the superpoint features, and the classification results $\widetilde{\mathbf{C}} \in \mathbb{Z}^q$ are derived by applying $\mathrm{argmax}$ operation over the similarity between the object queries and the text features

$$\widetilde{\mathbf{M}}^{\mathbf{3D}} \Leftarrow \mathbf{QS^{3D}}^\top > \tau, \widetilde{\mathbf{C}} \Leftarrow \mathrm{argmax}(\mathbf{QT}^\top, 1) \tag{4}$$

where $\tau$ is the threshold, $\mathrm{argmax}(\cdot, 1)$ returns the index of the maximum along the 1-th dimension.

However, since the annotations provided by OVSeg3R are partial view-level. The scene-level predictions need to be sent to the following View-wise Instance Partition (VIP) module to obtain the view-level predictions for supervision. (See Sec. 3.2)

## 3.2 View-wise Instance Partition

Given $q$ instance predictions from the 3D instance segmentator, including both the mask $\widetilde{\mathbf{M}}^{\mathbf{3D}}$ and classification $\widetilde{\mathbf{C}}$ predictions, we need to partition them to their corresponding views for supervision.

**Analysis.** Since the object queries are selected from the superpoints, each object query's content feature is initialized as its corresponding superpoint's feature. Therefore, the initial mask prediction tends to be the nearby superpoints of each object query. Meanwhile, in each decoder layer, when a query cross-attends to the superpoints, the attention is masked by the mask prediction from the last layer[1]. As a result, the multi-layer decoder operates as a form of iterative soft clustering, and the decoded mask of an object query typically corresponds to the entity that contains its corresponding superpoint. Therefore, if an object query's corresponding superpoint contains points reconstructed from the $v$-th view's pixels that describe an entity, then the instance mask decoded from the object query is most likely to correspond to that specific entity. Therefore, we explicitly assign the object query to the $v$-th view and truncate its scene-level mask prediction to retain only the region visible within the $v$-th view to obtain its view-level prediction in the $v$-th view. Without loss of generality, an entity here can refer to either a foreground object or background stuff in the scene.

**Design Details.** To implement the above VIP process, we first construct the view-belonging (visibility) mask for each superpoint $\mathbf{V^{sp}} \in \mathbb{B}^{V \times n}$ based on view-belonging mask of the reconstructed points $\mathbf{V^p} \in \mathbb{B}^{V \times N}$ and the superpoint mask $\mathbf{M^{sp}}$. For the $v$-th view, the process is formulated as

$$\mathbf{V}_v^{\mathbf{p}} \Leftarrow \left[ \overrightarrow{\mathcal{C}}(i)[0] \equiv v \right]_{i=1}^N, \mathbf{V}_v^{\mathbf{sp}} \Leftarrow \mathbf{M^{sp}}\mathbf{V}_v^{\mathbf{p}} > 0. \tag{5}$$

Then, according to $\mathbf{V}_v^{\mathbf{sp}}$, the view-belonging mask of object queries $\mathbf{V}_v^{\mathbf{q}} \in \mathbb{B}^q$ can be obtained by the slicing operation $\mathbf{V}_v^{\mathbf{q}} \Leftarrow \mathbf{V^{sp}}[\mathbf{q}]$. Finally, the partitioned mask predictions and class predictions for the $v$-th view, $\widetilde{\mathbf{m}}_v^{\mathbf{3D}} \in \mathbb{B}^{q_v \times HW}$[2] and $\widetilde{\mathbf{c}}_v \in \mathbb{Z}^{q_v}$ respectively, can be obtained by

$$\widetilde{\mathbf{m}}_v^{\mathbf{3D}} \Leftarrow \widetilde{\mathbf{M}}^{\mathbf{3D}}\left[\mathbf{V}_v^{\mathbf{q}}, \mathbf{V}_v^{\mathbf{sp}}\right], \widetilde{\mathbf{c}}_v \Leftarrow \widetilde{\mathbf{C}}\left[\mathbf{V}_v^{\mathbf{q}}\right], \tag{6}$$

where $q_v = \mathrm{Sum}(\mathbf{V}_v^{\mathbf{q}})$ is the number of queries that belong to the $v$-th view.

Although we must acknowledge that there is no mathematical foundation for ensuring every prediction is always assigned to the most appropriate view, the proposed strategy proves to be highly reliable in practice. By effectively reducing the risk of introducing incorrect false positives, it contributes to a stable training process. Our experimental results further confirm its effectiveness.

**Compatibility with Scene-level 3D Instance Segmentation Supervision**. It is worth noting that VIP is fully compatible with standard 3D instance segmentation supervision. For datasets that already provide scene-level annotations, there is no need to partition predictions to individual views for supervision; we can directly supervise the scene-level predictions using the scene-level annotations. This compatibility allows us to mix traditional costly annotated datasets, which provide manually adjusted reconstructed scenes and corresponding human annotations, with datasets that only have the raw videos, thereby enhancing the flexibility of the OVSeg3R training scheme. Moreover, for scene-level supervision, instead of performing a global matching between predictions and annotations via Hungarian matching (Carion et al., 2020), we follow the previous method (Kolodiazhnyi et al., 2024b; Qu et al., 2025) to sparsify the matching based on the relationship between the superpoints used to initialize object queries and the ground-truth masks. Specifically, a prediction can be matched to a ground-truth annotation only if the superpoint used to initialize its object query lies within that annotation's mask. This design strengthens the connection between the superpoint used for object query initialization and the final mask prediction, thereby consolidating the foundation of VIP from the perspective of supervision and improving its reliability.

---

[1] For the first layer, no reference mask is available, we do not restrict cross-attention via an attention mask.

[2] Here we assume all the pixels are correctly reconstructed and $HW = \mathrm{Sum}(\mathbf{V}_v^{\mathbf{p}})$ for notation simplicity.

### 3.3 2D INSTANCE BOUNDARY-AWARE SUPERPOINT

**Analysis.** Current methods typically employ the Felzenszwalb (Felzenszwalb & Huttenlocher, 2004), a graph-based segmentation algorithm, to over-segment large point sets into compact superpoints, reducing the following computational overhead. The algorithm constructs a graph by connecting each 3D point to its K-Nearest Neighbors (KNN). Subsequently, adjacent points in the graph that satisfy geometric continuity constraints will be clustered into the same superpoint.

However, as shown in Fig. 3 (b), the reconstruction results are often over-smoothed, leading to insufficient geometric distinction at instance boundaries. Simply constructing the superpoint graph with isotropic KNN, edges are inevitably formed between points that are geometrically continuous but belong to different instances. As a result, after the clustering in Felzenszwalb (see Sec A.2 in appendix), points from multiple instances, especially those lacking geometric salience, may be erroneously merged into the same superpoint, violating instance boundaries.

Therefore, we propose the IBSp to use the 2D instance masks to constrain the construction of superpoint graph, finally providing a better segmentation of superpoint.

**Design Details.** As described in Algorithm 1, after identifying the K-nearest neighbor (KNN) points for each 3D point, IBSp further projects the endpoints of each edge into 2D according to the 2D-3D correspondence $\mathcal{C}$ to obtain the corresponding instance indices from the 2D instance masks $\mathbf{M^{2D}}$. The edge is then deliberately disconnected if the two points do not belong to the same 2D instance. This pruned graph ensures that the points that belong to different instances are clustered into disconnected subgraphs, preventing inter-instance merging during the subsequent Felzenszwalb segmentation process.

---

**Algorithm 1** 2D Inst. Bound.-aware S.point Graph

**Require:** $\mathbf{P}, \mathcal{C}, \mathbf{M^{2D}}$
**Ensure:** Edges $\mathcal{E}$
1: Initialize $\mathcal{E} \leftarrow []$
2: **for** each point $i$ **do**
3:     Find neighbors: $\mathbf{K}_i \leftarrow \mathrm{KNN}(\mathbf{P}_i, \mathbf{P})$
4:     **for** each neighbor $j$ in $\mathbf{K}_i$ **do**
5:         $o_i \leftarrow \mathbf{M^{2D}}_{v_i,x_i,y_i}, (v_i, x_i, y_i) \leftarrow \overrightarrow{\mathcal{C}}(i)$
6:         $o_j \leftarrow \mathbf{M^{2D}}_{v_j,x_j,y_j}, (v_j, x_j, y_j) \leftarrow \overrightarrow{\mathcal{C}}(j)$
7:         **if** $o_i = o_j$ **then**
8:             Add edge $(i, j)$ to $\mathcal{E}$
9:         **end if**
10:     **end for**
11: **end for**
12: **return** $\mathcal{E}$

---

## 4 EXPERIMENTS

### 4.1 DATASETS

We validate the effectiveness of OVSeg3R based on ScanNetv2 (Dai et al., 2017) and ScanNet200 (Rozenberszki et al., 2022). They share the same 1,513 scenes, with 1,201 used for training and the remaining 312 for evaluation. The difference is that ScanNetv2 provides human-annotated instance masks for only 20 classes, whereas ScanNet200 extends the annotations to 200 classes. For each scene, both manually refined high-quality point clouds and the corresponding raw RGB videos are available. We reconstruct the raw videos using MASt3R-SLAM (Murai et al., 2025) and VGGT (Wang et al., 2025), resulting in reconstructed versions of the dataset, denoted as ScanNet3R-MSLAM and ScanNet3R-VGGT, respectively. Moreover, we leverage DINO-X to automatically produce view-wise open-vocabulary annotations for ScanNet3Rs as we have described in Sec 3.1.

### 4.2 EVALUATION SETTINGS

**Open Setting.** In this setting, the training data and annotations are not restricted. For example, some methods (Yang et al., 2023) are training free, some (Takmaz et al., 2023; Cao et al., 2025) are directly trained on the ScanNet200. For fair comparison, in this setting, we mix ScanNet200 and ScanNet3Rs for training. Following previous works, we report $\mathrm{mAP}_{25}$, $\mathrm{mAP}_{50}$, and mAP as the evaluation metrics. Specifically, $\mathrm{mAP}_{25}$ and $\mathrm{mAP}_{50}$ denote the mean Average Precision when the mask Intersection-over-Union (IoU) threshold is set to 25% and 50%, respectively, while mAP represents the average over multiple IoU thresholds ranging from 50% to 95% at a step of 5%. Moreover, to further analyze the comparison, the 200 classes are divided into head, common, and tail subsets according to their occurrence frequency, from high to low. We report the performance on the head and tail subsets separately, highlighting the class generalization ability among models.

**Standard Setting.** To further quantify the model's performance on novel categories, Open3DIS proposes using only the 20-class annotations provided by ScanNetv2 for supervision, while evaluating on all 200 classes in ScanNet200. Among these 200 classes, 50 are considered similar to the ScanNetv2 classes and are designated as base classes, while the remaining 150 classes, unseen in

Table 1: Comparison of OVSeg3R with prior methods on validation set of ScanNet200. Although SegDINO3D-VL supports open-vocabulary, when trained solely on ScanNet200, the limited annotation restricts it to closed-vocabulary, we denote it as SegDINO3D-VL directly. While, with OVSeg3R, SegDINO3D-VL is extended to open-vocabulary, we denote it as OVSeg3R.

| | All | | | Head | | | Tail | | |
|---|---|---|---|---|---|---|---|---|---|
| Method | mAP | $mAP_{50}$ | $mAP_{25}$ | mAP | $mAP_{50}$ | $mAP_{25}$ | mAP | $mAP_{50}$ | $mAP_{25}$ |
| *Closed-vocabulary* | | | | | | | | | |
| Mask3D | 27.4 | 37.0 | 42.3 | 40.3 | 55.0 | 62.2 | 18.2 | 23.2 | 27.0 |
| MAFT | 29.2 | 38.2 | 43.3 | - | - | - | - | - | - |
| OneFormer3D | 30.2 | 40.9 | 44.6 | 42.0 | 57.7 | 63.9 | 20.1 | 26.6 | 27.7 |
| ODIN | 31.5 | 45.3 | 53.1 | 37.5 | 54.2 | 66.1 | 24.1 | 36.6 | 41.2 |
| SegDINO3D | 39.8 | 52.1 | 58.6 | 46.0 | 63.2 | 71.5 | 36.2 | 44.9 | 51.0 |
| SegDINO3D-VL | 38.4 | 50.2 | 55.6 | 45.3 | 62.0 | 69.6 | 34.0 | 43.4 | 47.3 |
| *Open-vocabulary* | | | | | | | | | |
| SAM3D | 9.8 | 15.2 | 20.7 | 9.2 | - | - | 12.3 | - | - |
| SAI3D | 12.7 | 18.8 | 24.1 | 12.1 | - | - | 16.2 | - | - |
| SAM2Object | 13.3 | 19.0 | 23.8 | - | - | - | - | - | - |
| OpenMask3D | 15.4 | 19.9 | 23.1 | - | - | - | - | - | - |
| Open3DIS | 23.7 | 29.4 | 32.8 | 27.8 | - | - | 21.8 | - | - |
| Open-YOLO 3D | 24.7 | 31.7 | 36.2 | 27.8 | - | - | 21.6 | - | - |
| Any3DIS | 25.8 | - | - | 27.4 | - | - | 26.4 | - | - |
| LIFT-GS | 25.7 | 35.0 | 40.2 | - | - | - | - | - | - |
| OVSeg3R (Ours) | **40.7** | **53.0** | **59.5** | **44.6** | **61.1** | **68.8** | **42.7** | **53.1** | **58.7** |

ScanNetv2, are treated as novel classes. Thus, under this setting, we train on a mixture of ScanNetv2 and ScanNet3Rs for fair comparison. We report the models' mAP separately on the novel and base classes to demonstrate models' generalization ability.

### 4.3 IMPLEMENTATION DETAIL

To validate the effectiveness of OVSeg3R, we start by modifying the current state-of-the-art closed-vocabulary instance segmentator SegDINO3D. Specifically, its limited classification head is replaced with a similarity-based module that compares object features with text embeddings. Since the text encoder can accept arbitrary textual input, the classification is naturally extended to support the open-vocabulary setting. We denote this extended model as SegDINO3D-VL. We adopt CLIP as our text encoder. In the benchmark experiments, we leverage both ScanNet3R-MSLAM and ScanNet3R-VGGT to provide richer training data and achieve better overall performance. In the ablation studies, if not explicitly stated, we use only ScanNet3R-MSLAM by default.

### 4.4 COMPARISON WITH STATE-OF-THE-ARTS

**Open Setting**. To quantify the gains from the additional training data and annotations provided by OVSeg3R, after extending SegDINO3D to SegDINO3D-VL, we first train it under the traditional training scheme, using only ScanNet200. Since the annotations at this setting cover only 200 categories, the resulting model remains closed-vocabulary. As shown in Table 1, under this setting, we observe a performance drop. We attribute this to the additional model capacity required for aligning text and visual features, which can also be observed in 2D models (Liu et al., 2024b). Moreover, although the model in principle supports open-vocabulary generalization, the limited diversity of training data constrains its semantic generalization ability. Consequently, its performance on tail classes remains significantly lower than that on head classes. As a comparison, as shown in the last row of Table 1, when SegDINO3D-VL is trained under the proposed OVSeg3R training scheme with open-vocabulary annotations, the model not only gains open-vocabulary capability but also achieves a notable overall improvement (+2.3 mAP), outperforming even the closed-vocabulary methods. The detailed performance on the tail and head categories reveals the source of the improvement. Benefiting from stronger class generalization ability, the model achieves a significant gain on tail classes (+8.7 mAP), reducing the performance gap with head classes from -11.3 mAP to -1.9 mAP.

**Standard Setting**. As shown in Table 2, under this setting, our model achieves state-of-the-art overall performance. Importantly, this SoTA result is not due to the minor improvement on base classes (about +0.0 mAP), but largely stems from the enhanced generalization to novel classes, where our method surpasses previous approaches by approximately +7.7 mAP, further demonstrating the effectiveness of OVSeg3R.

Table 2: Comparison of OVSeg3R with prior methods on the standard setting. $mAP_n$ and $mAP_b$ indicate the mAP performance on novel and base classes.

| Method | mAP | $mAP_n$ | $mAP_b$ |
|---|---|---|---|
| PLA | 4.5 | 0.3 | 15.8 |
| OpenScene+Mask3D | 8.5 | 7.6 | 11.1 |
| OpenMask3D | 12.6 | 11.9 | 14.3 |
| Open3DIS | 19.0 | 16.5 | 25.8 |
| Any3DIS (SAM2-L) | 19.1 | - | - |
| OVSeg3R (Ours) | **24.6** | **24.2** | **25.8** |

Table 3: Ablation studies on the designs for stable training and the reconstruction data. S.3R-M and S.3R-V stand for ScanNet3R-MSLAM and ScanNet3R-VGGT respectively.

| VIP | IBSp | S.3R-M | S.3R-V | mAP | $mAP_n$ |
|---|---|---|---|---|---|
| ✗ | ✓ | 100% | 0% | 18.4 | 16.2 |
| ✓ | ✗ | 100% | 0% | 23.6 | 22.0 |
| ✓ | ✓ | 100% | 0% | 23.9 | 23.0 |
| ✓ | ✓ | 0% | 0% | 5.0 | 4e-4 |
| ✓ | ✓ | 1% | 0% | 7.0 | 3.3 |
| ✓ | ✓ | 10% | 0% | 16.8 | 14.8 |
| ✓ | ✓ | 100% | 100% | **24.6** | **24.2** |

## 4.5 ABLATIONS

To clarify the impact of our designs, we conduct ablation studies under the standard setting.

**Ablation on Designs for Stable Training**. As shown in Table 3, removing View-wise Instance Partition (VIP) introduces numerous false positives, leading to a substantial performance drop, which underscores its importance for stable training. Meanwhile, replacing IBSp with a geometric-only superpoint also causes a performance degradation, but relatively smaller. This indicates that VIP plays a more prominent role in stabilizing the training. However, this does not imply that IBSp is unimportant. In practical scenarios, where point clouds are typically reconstructed from video rather than manually refined as in the evaluation set, IBSp remains crucial to ensure that the geometrically less salient object can be segmented out.

**Ablation on Reconstruction Data**. As shown in Table 3, when only a subset of scenes from ScanNetv2 is used to construct the reconstructed data for OVSeg3R's open-vocabulary training, the model performance drops noticeably as the data volume decreases to 10% and 1%. In particular, when the data volume reaches 0%, the training essentially degenerates to the traditional scheme. Without any open-vocabulary supervision provided by OVSeg3R, the model's performance on novel classes is nearly zero. Moreover, when using two different 3D reconstructors (MASt3R-SLAM and VGGT) to provide data for OVSeg3R, the model achieves even better performance, despite both datasets being reconstructed from the same scenes and sharing the same annotations provided by the 2D model. This suggests that point clouds generated by different 3D reconstructors can be regarded as a form of input data augmentation.

## 5 VISUALIZATION

To intuitively show OVSeg3R's open-vocabulary segmentation ability and its robustness to input point clouds, we perform segmentation on the reconstruction of an in-the-wild video with the text prompt 'tripod . power strip .'. Here, 'tripod' is a novel category that is not included in existing datasets, while 'power strip' is a long-tail category. As shown in Fig. 4, one tripod and two power strips are correctly found and segmented out. See appendix and supplementary material for more visualization and the corresponding original video and 3D segmentation results.

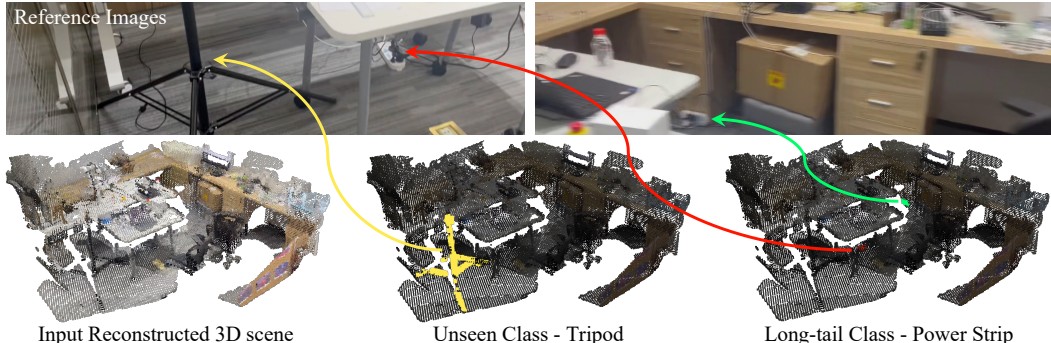

Figure 4: Visualization of segmentation results of OVSeg3R on in-the-wild data. We provide the frames in which each object is most clearly visible in the video as references.

## 6 CONCLUSION

In this paper, we have presented OVSeg3R, a novel training scheme for open-vocabulary 3D instance segmentation. By fully leveraging the modern 3D reconstruction and well-studied 2D instance segmentation models, OVSeg3R enables learning of open-vocabulary 3D instance segmentation, improving the models' native 3D perception ability. The proposed designs, View-wise Instance Partition and 2D Instance Boundary-aware Superpoint, enhance the stability of the training scheme. With these designs, OVSeg3R extends the state-of-the-art closed-vocabulary to open-vocabulary. The strong class generalization brought by OVSeg3R not only substantially reduces the performance gap between head and tail classes, but also leads to consistent improvements in both open and standard settings, demonstrating the effectiveness of OVSeg3R.

## ETHICS STATEMENT

This work focuses on developing methods for open-vocabulary 3D instance segmentation, aiming to improve the perception and understanding of complex 3D environments. Our study does not involve human subjects, personally identifiable information, or sensitive data. All datasets used are publicly available and widely adopted in the community. While open-vocabulary recognition carries potential risks of misuse, such as unintended surveillance or biased predictions, we emphasize that our approach is designed for research purposes and should be applied responsibly. We encourage future use of this technology to adhere to ethical guidelines, avoid privacy violations, and mitigate potential biases in downstream applications.

## REPRODUCIBILITY STATEMENT

We have made every effort to ensure the reproducibility of OVSeg3R. Detailed descriptions of the implementation, training data and evaluation settings are provided in Sec. 4.3, Sec. 4.1 and Sec. 4.2 respectively. The source code will be released upon acceptance, enabling researchers to replicate and extend our results.

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

# A  APPENDIX

## A.1  MORE ABLATIONS

Table 4: Ablation on the quality of 3D reconstruction and the input 2D feature.

| Filter out Fail Recon. | Universal DINO-X Feat. | mAP | $\text{mAP}_n$ | $\text{mAP}_b$ |
|---|---|---|---|---|
| ✓ | ✓ | 23.9 | 23.0 | 26.2 |
| ✗ | ✓ | 23.0 | 22.3 | 25.1 |
| ✓ | ✗ | 16.7 | 14.0 | 23.9 |

**Ablation on 3D Reconstruction Quality** Although 3D reconstruction methods have advanced considerably (Wang et al., 2024; Leroy et al., 2024; Murai et al., 2025; Wang et al., 2025), failures still occur due to errors in camera parameters or depth estimation. To assess their impact on the model, we keep the failed reconstructions and conduct an ablation study. As shown in Table 4, filtering out failed reconstructions yields a clear performance gain, demonstrating the potential of OVSeg3R. With continued advances in 3D reconstruction, as accuracy improves and failure rates decrease, the contribution of OVSeg3R in the 3D scene understanding will be further enhanced.

**Ablation on Input 2D Features** Since our experiments are based on SegDINO3D-VL, an extension of SegDINO3D (Qu et al., 2025), which proposes to leverage well-trained 2D features to help data-hungry 3D models in understanding 3D scenes. Thus, as in SegDINO3D, input 2D image- and object-level features are important. By default, we use DINO-X (Ren et al., 2024) in universal mode[3] to provide features. For comparison, we also extract features using DINO-X in the regular mode, restricting text prompts to the 20 ScanNet classes. As shown in Table 4, limiting the 2D model's attention to these 20 classes prevents it from providing sufficient information for open-vocabulary recognition. While this has little negative impact on base classes (-1.2 mAP), it leads to a substantial performance drop on novel classes (-8.3 mAP).

## A.2  CONSTRUCTING SUPERPOINT WITH FELZENSZWALB ALGORITHM

After obtaining the 2D instance boundary constrained superpoint graph edges $\mathcal{E}$ in Sec 3.3, we apply the Felzenszwalb segmentation algorithm to generate superpoints. The algorithm employs a disjoint-set forest data structure to efficiently manage connected components and uses an adaptive threshold mechanism to control the granularity of segmentation. To evaluate the geometric continuity, we need to pre-calculate the vertex normal $\mathbf{N} \in \mathbb{R}^{N \times 3}$ for each point in $\mathbf{P}$. We follow previous methods to use the Principal Component Analysis (PCA) on each point's local K-nearest-neighbor points, and select the eigenvector corresponding to the smallest eigenvalue as the vertex normal. Given the maximum tolerance threshold $Sp_{thresh} \in \mathbb{R}^+$ and the minimum superpoint size $Sp_{min} \in \mathbb{Z}^+$, Algorithm 2 produces the superpoint mask $\mathbf{M^{sp}}$.

---

**Algorithm 2** Superpoint Construction via Felzenszwalb Algorithm

---

**Require:** Point cloud $\mathbf{P} \in \mathbb{R}^{N \times 3}$, normals $\mathbf{N} \in \mathbb{R}^{N \times 3}$, edges $\mathcal{E}$ from Algorithm 1
**Require:** Threshold parameter $Sp_{thresh} \in \mathbb{R}^+$, minimum segment size $Sp_{min} \in \mathbb{Z}^+$
**Ensure:** Superpoint mask $\mathbf{M^{sp}}$
 1: Initialize disjoint-set forest $\mathcal{U}$ with $N$ singleton components
 2: Initialize edge weights $W \leftarrow \{\}$
 3: Initialize adaptive thresholds $\mathbf{t}[i] \leftarrow Sp_{thresh}$ for $i = 1, \dots, N$
 4: Initialize superpoint labels $\mathbf{Sp}[i] \leftarrow i$ for $i = 1, \dots, N$
    {Compute edge weights based on geometric continuity}
 5: **for** each edge $(i, j) \in \mathcal{E}$ **do**
 6:    $dot \leftarrow \mathbf{N}_i \cdot \mathbf{N}_j$ {Normal similarity}
 7:    $w \leftarrow 1 - dot$ {Base weight from normal difference}
 8:    $W[(i, j)] \leftarrow w$
 9: **end for**

---

[3]In the universal mode of DINO-X, users do not need to provide text prompts specifying target classes, DINO-X automatically detects all objects.

---

**Algorithm 2** Superpoint Construction via Felzenszwalb Algorithm

---

10: Sort edges in $\mathcal{E}$ by increasing weight: $W[e_1] \leq W[e_2] \leq \ldots \leq W[|\mathcal{E}|]$
   {Felzenszwalb graph-based segmentation with adaptive thresholds}
11: **for** each edge $e_k = (i, j)$ in sorted order **do**
12:   $root_i \leftarrow \mathcal{U}.\text{find}(i)$ {Find root of component containing $i$}
13:   $root_j \leftarrow \mathcal{U}.\text{find}(j)$ {Find root of component containing $j$}
14:   **if** $root_i \neq root_j$ **and** $W[e_k] \leq \mathbf{t}[root_i]$ **and** $W[e_k] \leq \mathbf{t}[root_j]$ **then**
15:     $\mathcal{U}.\text{union}(root_i, root_j)$ {Merge components}
16:     $new_{root} \leftarrow \mathcal{U}.\text{find}(root_i)$ {Get merged component root}
17:     $\mathbf{t}[new_{root}] \leftarrow W[e_k] + \frac{Sp_{thresh}}{|\mathcal{U}.\text{size}(new_{root})|}$ {Update adaptive threshold}
18:   **end if**
19: **end for**
   {Post-processing: merge small segments}
20: **for** each edge $e_k = (i, j)$ in $\mathcal{E}$ **do**
21:   $root_i \leftarrow \mathcal{U}.\text{find}(i)$
22:   $root_j \leftarrow \mathcal{U}.\text{find}(j)$
23:   **if** $root_i \neq root_j$ **and** $(|\mathcal{U}.\text{size}(root_i)| < Sp_{min}$ **or** $|\mathcal{U}.\text{size}(root_j)| < Sp_{min})$ **then**
24:     $\mathcal{U}.\text{union}(root_i, root_j)$ {Force merge small segments}
25:   **end if**
26: **end for**
   {Extract final superpoint labels}
27: **for** $i = 1$ to $N$ **do**
28:   $\mathbf{Sp}[i] \leftarrow \mathcal{U}.\text{find}(i)$
29: **end for**
30: Relabel $\mathbf{Sp}$ to consecutive indices starting from 0
31: $\mathbf{M^{sp}} \leftarrow \text{OneHot}(\mathbf{Sp})^{\top}$
32: **return** $\mathbf{M^{sp}}$

---

### A.3 MORE VISUALIZATIONS

To further demonstrate the superiority of OVSeg3R and its potential in downstream applications, such as robotic navigation, manipulation and video understanding, we present additional visualizations of predictions on out-of-distribution in-the-wild data. For the segmentation targets, as shown in Fig. 5, Fig. 6 and Fig. 7, rather than focusing on the geometrically salient furniture objects that dominate existing datasets, we highlight the model's performance on tail, unseen, and geometrically non-salient objects.

The original video, reconstructed scene, and the segmentation results are provided in our supplementary material for better visualization.

### A.4 LLM USAGE

We use LLMs to polish the writing of this paper, mainly for correcting grammatical errors and improving readability. In addition, LLMs are used to assist in analyzing potential ethical considerations of this work.

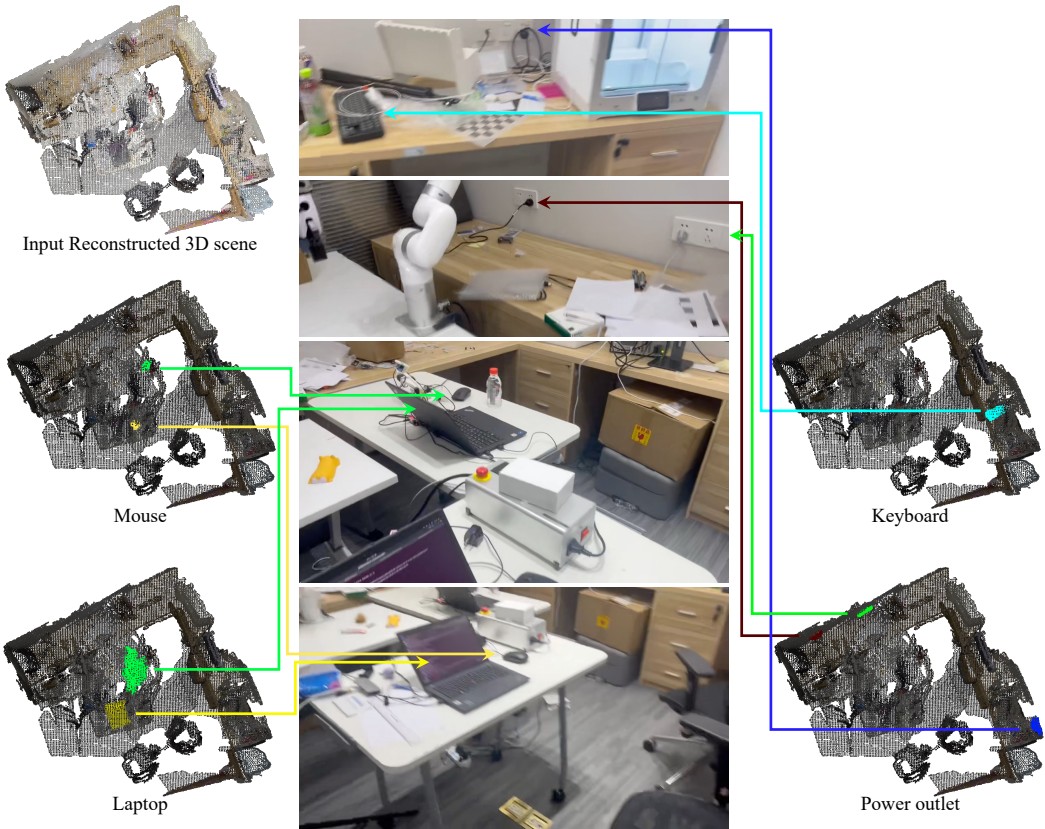

Figure 5: Input text prompt: "laptop . mouse . keyboard . power outlet .". Although the power outlet, keyboard, and mouse are not geometrically salient, making them difficult to identify even for humans in the reconstructed 3D point clouds, OVSeg3R can still accurately locate and segment them. For the laptop case, despite local reconstruction failures caused by inaccurate camera parameter estimation during reconstruction, OVSeg3R is still able to segment it (with green mask). Best viewed in the electronic version or by referring to the 3D segmentation results provided in the supplementary material.

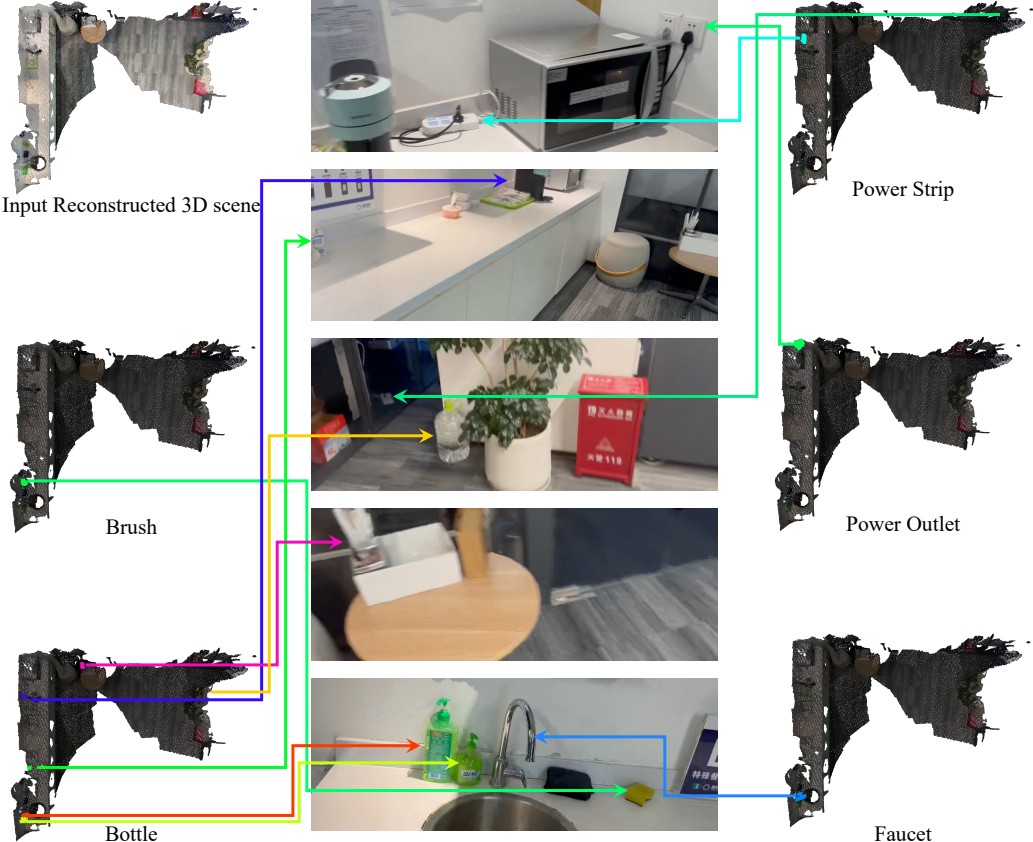

Figure 6: Input text prompt: "bottle . brush . faucet . power outlet . power strip .". Despite the two bottles near the faucet being small (compared with the furniture objects that dominate existing datasets) and closely positioned, our model can still segment and distinguish them. Moreover, although the 'brush' class is not present in existing datasets, OVSeg3R is still capable of recognizing and segmenting it. Best viewed in the electronic version or by referring to the 3D segmentation results provided in the supplementary material.

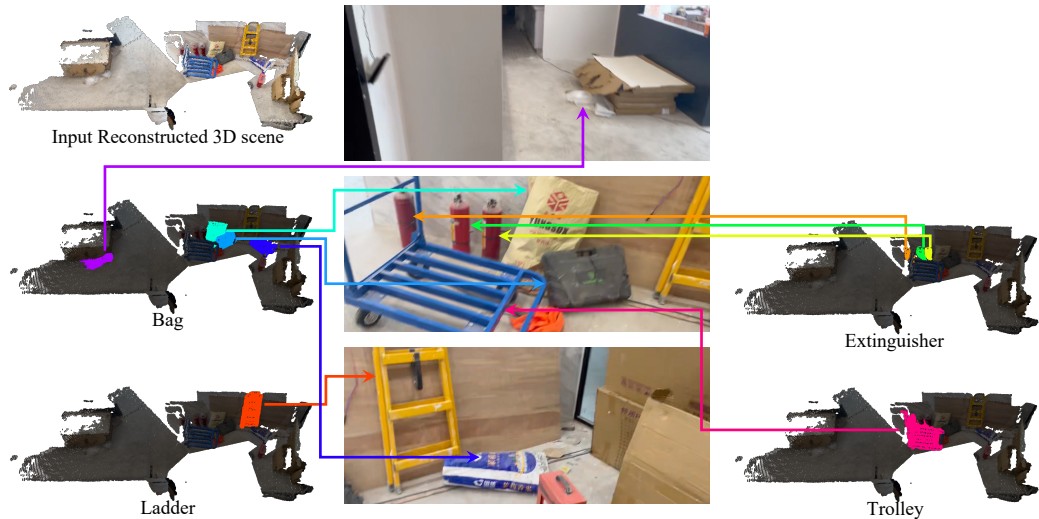

Figure 7: Input text prompt: "bag . ladder . extinguisher . trolley .". Despite the white plastic bag (with the purple mask) blending into the floor and forming strong geometric continuity, our 2D Instance-Boundary-aware Superpoint (IBSp) enables OVSeg3R to successfully segment it out. Moreover, although the 'trolley' class is not present in existing datasets, OVSeg3R is still capable of recognizing and segmenting it. Best viewed in the electronic version or by referring to the 3D segmentation results provided in the supplementary material.

