# OpenReview forum: "OVSeg3R: Learn Open-vocabulary Instance Segmentation from 2D via 3D Reconstruction"
_ICLR.cc/2026/Conference — ICLR 2026 Poster_

### Official Review · Reviewer_3KhS · 2025-10-27

**Soundness:** 2
**Presentation:** 2
**Contribution:** 2
**Rating:** 4
**Confidence:** 3

**Summary:**

The paper presents OVSeg3R, a training scheme for open-vocabulary 3D instance segmentation that avoids manual 3D instance annotations. Instead, it uses two readily available sources: (1) video-based 3D reconstruction to obtain a point cloud and 2D–3D correspondences, and (2) strong open-vocabulary 2D instance segmentation to get per-frame instance masks and category names. These 2D masks are lifted into 3D and used as supervision.

To make this supervision stable, the authors introduce (i) View-wise Instance Partition (VIP), which matches each predicted 3D instance only to the visible portion in each view to avoid penalizing regions that were never observed, and (ii) 2D Instance Boundary-aware Superpoints (IBSp), which uses 2D instance boundaries to prevent small or low-relief objects from being merged into large planar superpoints.

They apply OVSeg3R to a strong recent 3D instance segmentation backbone (SegDINO3D), modifying only the classification head to score similarity with text embeddings for open-vocabulary recognition. Trained under OVSeg3R, this model outperforms both prior open-vocabulary 3D methods and a traditionally trained baseline, with especially large gains on long-tail and novel categories.

**Strengths:**

- Clear motivation. The paper tackles the real bottleneck for open-vocabulary 3D instance segmentation: 3D instance masks with textual labels are expensive to annotate, but videos + strong 2D open-vocabulary segmentors + modern 3D reconstruction are becoming cheap and ubiquitous. OVSeg3R leverages exactly those ingredients.
- Significant gains on long-tail and novel categories. The method not only improves overall mAP but also drastically narrows the head–tail gap and boosts novel categories in the standard open-vocabulary setting. This indicates real progress toward robust, open-world 3D perception rather than just re-labeling known classes.
- Benchmark thoroughness. The paper evaluates both in an open setting (mixing ScanNet200 with reconstructed ScanNet3R-* scenes) and in the widely used standard setting (only ScanNetv2’s 20 labeled classes for supervision, evaluate on all 200 classes and report base vs. novel performance). The setting of these benchmarks is sufficient.

**Weaknesses:**

- Generalization to other backbones is not empirically shown. OVSeg3R is advertised as a training scheme, but all quantitative results are on a single architecture family (SegDINO3D to SegDINO3D-VL). The author does not display the results of Mask3D, OneFormer3D, ODIN, etc. retrained under OVSeg3R. This makes it hard to judge whether VIP and IBSp are truly architecture-agnostic or mainly tailored to SegDINO3D’s query-from-superpoints decoder.
- Entangled factors in the main comparison. The baseline SegDINO3D-VL trained in the traditional way uses only ScanNet200’s manually aligned 3D annotations. The OVSeg3R model, in contrast, gets access to reconstructed ScanNet3R-* scenes plus lifted 2D pseudo-labels with potentially more and diversified supervision. It is therefore non-trivial to isolate how much of the gain comes from the proposed VIP/IBSp training mechanism itself vs. simply having more data. An ablation experiment that feeds SegDINO3D-VL the same data volume but without VIP/IBSp would clarify causality.
- Presentation clarity. Because the paper names the final model “OVSeg3R (Ours),” it can be misunderstood as a new architecture, rather than “SegDINO3D-VL trained with our pseudo-labeling scheme (OVSeg3R) plus VIP+IBSp.” This blurs the line between the backbone and the training recipe, and makes it trickier to interpret fairness in the comparisons.

**Questions:**

These are the major concerns that will be related to my final rating:
- Backbone generality. Can you demonstrate OVSeg3R on other closed-vocabulary 3D instance segmentation backbones (e.g., Mask3D, OneFormer3D, ODIN)? Even a smaller-scale experiment would help establish that VIP and IBSp are not tightly coupled to SegDINO3D’s query-from-superpoints architecture. If not, please discuss what specific assumptions OVSeg3R makes about the backbone.
- Ablation disentangling data volume vs. VIP/IBSp. In Table 1 / Table 3 comparisons, OVSeg3R benefits from additional *ScanNet3R-** reconstructed scenes and thus more pseudo-labels. Could you run a control where SegDINO3D-VL is exposed to the same reconstructed scenes and text-prompts but without VIP or IBSp?

And here are some minor questions that may not influence the rating:
- Can you further clarify how to reproduce your visualization with your ply files? It seems I need to manually adjust the scale of each point cloud after opening it in MeshLab.
- Metrics beyond AP. Since the main story is *we drastically improve novel/long-tail categories*, can you report recalls on head / common / tail categories, or at least qualitative failure modes for novel classes? This would make the open-vocabulary claim even more convincing.

---

> ### Author Response · Authors · 2025-11-23
> **Rebuttal**
>
> ## Are VIP and IBSp architecture-agnostic?
> For the Instance Boundary-aware Superpoint (IBSp), it is motivated by our observation of the over-smoothed reconstruction results. Any method that operates on reconstructed 3D scenes, regardless of the architecture or supervision, can benefit from IBSp.
>
> For View-wise Instance Partition (VIP), our current implementation method is to obtain each object query's view-belonging mask through each superpoint's view-belonging mask. This admittedly couples VIP to the query-from-superpoints architecture. However,
> 1. This coupling is only limited at the implementation level. The contribution of our proposed core insight of VIP, partitioning instance predictions across different views to avoid excessive false negatives and stabilize pseudo-label learning, should not be ignored.
> 2. Moreover, the query-from-superpoints architecture is currently the dominant and most effective design for 3D instance segmentation. In fact, this architecture was exactly the improvement over Mask3D [1] originally introduced by Oneformer3D [2], which the reviewer mentioned. Most of the subsequent works, including the SegDINO3D that we follow, have all adopted this design.
> More importantly, the query-from-superpoints architecture was inspired by the query-selection [3] approach used in all mainstream 2D detectors. Its effectiveness has already been widely recognized in the 2D perception domain [4,5,6].
>
>
> ## Ablation disentangling data volume vs. VIP/IBSp.
>
> As suggested by the reviewer, we conducted an experiment using the same data but without VIP and IBSp, in order to verify that the performance improvement of the model is not solely due to the additional pseudo labels provided by more data. Due to time and resource constraints, this experiment has currently only run for half of the iterations. We will complete the experiment and include the results in the ablation study (to be inserted as the first row of Table 3).
>
> | Experiment     | mAP  |
> | -------------  | ---- |
> | w/o VIP & IBSp | 15.1 |
> | w VIP & IBSp   | 20.8 |
>
> ## Recalls on head / common / tail classes.
> Due to space limitations in the table of the main paper, we only reported the APs performance for the head and tail subsets. As suggested by the reviewer, here we also present the recall for the head, common, and tail subsets. As shown below, consistent with the trends observed in the overall metric (APs), OVSeg3R demonstrates consistent superiority in recall as well.
>
> | Methods      | Head | Common | Tail |
> | -------------| ---- | ------ | ---- |
> | SegDINO3D-VL | 68.5 | 58.2   | 61.1 |
> | OVSeg3R      | 70.0 | 64.3   | 70.1 |
> | Improvement  | +1.5 | +6.1   | +9.0 |
>
>
> ## Reproduction of the visualization.
>
> To reproduce our visualization results, it is sufficient to import all .ply files (including the reconstructed full scene and the segmentation results of individual objects) into a single MeshLab project. Simply open MeshLab, select all .ply files in the directory, and drag them into the MeshLab GUI. There is no need to manually adjust the scale of each point cloud (.ply).
>
> ## Reference
> [1] Mask3D: Mask Transformer for 3D Semantic Instance Segmentation
>
> [2] Oneformer3d: One Transformer for Unified Point Cloud Segmentation
>
> [3] Deformable DETR: Deformable Transformers for End-to-end Object Detection
>
> [4] DINO: DETR with Improved DeNoising Anchor Boxes for End-to-End Object Detection
>
> [5] Grounding DINO: Marrying DINO with Grounded Pre-Training for Open-Set Object Detection
>
> [6] DINO-X: A Unified Vision Model for Open-World Object Detection and Understanding

---

> ### Comment · Reviewer_3KhS · 2025-11-24
>
> Thank you for the rebuttal. After reading the other reviews and the authors’ responses, I believe my concerns have been addressed, and I will raise my rating to 6.

---

> > ### Author Response · Authors · 2025-11-24
> > **Acknowledgment to Reviewer 3Khs**
> >
> > We sincerely thank the reviewer for the constructive feedback and for raising the rating. We are glad that our responses were helpful in addressing the concerns. We will incorporate the suggested clarifications and improvements in the camera-ready version.

---

### Official Review · Reviewer_G6nS · 2025-10-28

**Soundness:** 3
**Presentation:** 1
**Contribution:** 2
**Rating:** 6
**Confidence:** 3

**Summary:**

This paper presents OVSeg3R, a framework designed to tackle open-vocabulary 3D instance segmentation by leveraging the capabilities of existing 2D open-vocabulary segmentation models and 3D reconstruction models. The core idea is to use 3D reconstruction from videos to lift 2D instance predictions into the 3D space, generating large-scale, automatic supervision for a 3D segmentation network. The authors introduce two main technical contributions to stabilize this 2D-to-3D lifting process: 1) the View-wise Instance Partition (VIP) algorithm to partition the predicted 3D segmentation masks to their respective views for training supervision, and 2) the 2D Instance Boundary-aware Superpoint (IBSp) clustering, which leverages 2D mask boundaries to directly constrain 3D superpoint generation, thereby preventing over-smoothing and ensuring sufficient geometric distinction at instance boundaries that is often lost in 3D reconstruction outputs. The proposed method demonstrates strong performance, successfully extending closed-vocabulary 3D segmentation to the open-vocabulary regime.

**Strengths:**

1. **Addressing a Critical Problem**: Open-vocabulary 3D segmentation is a crucial area for real-world applications (robotics, AR/VR), and this paper offers a scalable, data-efficient solution by exploiting available 2D models and 3D reconstruction pipelines.

2. **Innovative Supervision Strategy**: The automatic generation of 3D supervision by projecting 2D open-vocabulary masks is a practical and creative approach to circumvent the prohibitive cost of manual 3D annotation.

3. **Targeted Solutions**: The paper correctly identifies the two main issues of 2D-to-3D lifting—inconsistency/partiality in view projections and over-smoothing of 3D geometry—and proposes two respective solutions: VIP and IBSp, which effectively use 2D annotations from 2D segmentation models to refine and stabilize the 3D pseudo-labeling and feature grouping process.

4. **Competitive Results**: The reported performance metrics suggest that OVSeg3R is an effective way to transfer knowledge from the 2D domain to 3D, achieving state-of-the-art results in the open-vocabulary 3D segmentation setting.

**Weaknesses:**

1. **Clarity of Technical Details**: The description in lines 272-278 is confusing: concepts like "the attention is masked by the mask prediction of the previous layer", how "the multi-layer decoder functions like K-means clustering" to associate a superpoint with an entity, and "pixels that describe an entity" are expressed vaguely and require rephrasing for clarity. In addition, it is better to add captions for math symbols in Fig. 2

2. **Reliance on a Single 2D Perception Model**: While the authors test the framework's robustness across different 3D reconstruction models (e.g., VGGT and MASt3R-SLAM), the core pipeline relies on a single 2D open-vocabulary segmentation model (DINO-X). Given that the quality of the 3D pseudo-labels is highly dependent on the 2D mask predictions and their boundaries, a sensitivity analysis or ablation study comparing results across different 2D perception models (e.g., models with different architectures or mask qualities) is missing. This limits the understanding of OVSeg3R's generalization capability beyond the specific chosen 2D backbone.

3. **Limited Generalizability Testing**: The evaluation is conducted primarily on a single dataset, ScanNetV2. While the methodology is aimed at open-vocabulary generalization, testing its performance, robustness, and calibration on diverse 3D datasets (e.g., matterport3d, sunrgbd, or datasets with different scene characteristics, density, and scales) is crucial to verify generalizability.

**Questions:**

1. **Domain Discrepancy (Train vs. Test Data)**: The method is trained on point clouds generated by 3D reconstruction models. During inference, point clouds may be obtained from different sensors (like LiDAR or consumer depth cameras), which produce data with different noise characteristics, density, and quality than reconstructed scenes. Does this discrepancy between the training data distribution (reconstructed) and potential inference data distribution (direct sensor capture) hinder the model's performance?

2. **Failure Modes and Error Analysis**: Since this is a pseudo-labeling approach, what are the observed failure modes for OVSeg3R? Does the VIP mechanism also introduce a significant number of false negatives?

---

> ### Author Response · Authors · 2025-11-23
> **Rebuttal**
>
> ## Domain discrepancy between training and evaluation.
>
>
> In practice, beyond the inherent noise of reconstructed point clouds, our training pipeline, following prior works, applies extensive data augmentations to the input point clouds, including random elastic deformation, rotation, and translation, etc. These augmentations make the model highly robust to various forms of geometric noise. Moreover, since the backbone operates on a voxelized representation, variations in point cloud density have a limited impact on the final predictions.
>
> Meanwhile, from our empirical observations, the model’s performance consistently improves as the quality of the input point cloud improves; it does not degrade when the input is less noisy than the points seen during training. Considering that 3D data captured from depth cameras or LiDAR typically have higher geometric fidelity than reconstructions obtained purely from RGB inputs, the model is expected to perform at least as well on such point clouds obtained from 3D sensors.
>
> The strong performance achieved on the high-quality point clouds of the evaluation set (no noise can also be regarded as one kind of noise pattern) further demonstrates the robustness of our model to variations in input quality.
>
> ## Failure modes and error analysis in VIP.
> Failure mode: When the reconstruction algorithm yields highly inaccurate estimates of camera extrinsics, the reconstructed scene may collapse into heavy noise, preventing our model from performing well, due to the large domain gap between the training and test distributions. In practice, this issue still occurs even when using the state-of-the-art reconstruction methods (e.g., VGGT / MASt3R-SLAM). However, the proportion of such severely corrupted reconstructions is relatively small and has limited impact on overall performance.
>
> Error analysis in VIP: Indeed, while VIP helps reduce false positives, it can also introduce occasional false negatives. However, this typically occurs when the positive prediction itself has a low-quality mask, causing its subset in the current sub-scene to have a higher matching cost with the ground-truth subset than some negative predictions. Such cases are rare and mostly occur in the early stages of training. As training progresses, the likelihood of these occurrences decreases rapidly. Moreover, during early training, these sporadic noises are negligible compared to the overall instability at initialization. Consequently, these occasional errors do not destabilize training. On the contrary, by removing a large number of false positives, VIP enables OVSeg3R to train stably on pseudo-labels, with the training loss consistently decreasing. The significant performance improvement provided by VIP in Table 3 verifies this.
>
> ## Clarity of technical details.
>
> We thank the reviewer for pointing out the ambiguity in the description. We will revise this paragraph in the camera-ready version to provide a clearer explanation. We will also add captions for mathematical symbols in Fig. 2 for completeness and clarity.
>
>
> ## OVSeg3R's generalization ability with different mask qualities.
>
> We inject controlled 2D mask noise with different kernel sizes for random dilation and erosion, and use Grounded SAM [1], as a replacement of DINO-X to provide 2D masks with noisy class labels. Limited by the time and resources, we conduct the experience with only about half of the iterations. The results are shown below:
>
> | Noise | mAP |
> | ----------------- | --- |
> | No Noise                      | 34.6 |
> | 0~10 pixels Noise Kernal Size | 34.6 |
> | 0~20 pixels Noise Kernal Size | 33.9 |
> | 0~40 pixels Noise Kernal Size | 33.4 |
> | Grounded SAM Masks            | 32.8 |
>
> We observe that when a small amount of noise is added to the mask boundaries (0–10 pixels), the performance of the model remains nearly unaffected. As the noise level increases further, the performance does degrade, but the drop remains within an acceptable range (-1.2 mAP).
>
> When we replace DINO-X with Grounded SAM as the 2D mask provider, the mask quality deteriorates for a different reason. Grounded SAM generates masks from the 2D bounding boxes predicted by GroundingDINO, and the produced masks do not always correspond to the exact objects intended by the boxes (one 2D box may contains multiple objects, SAM does not know which specific object within the box to segment). This misalignment results in incorrect class labels for some masks, which directly harms the open-vocabulary classification ability of the model. As shown in our results, this causes a performance drop of approximately 1.8 mAP.
>
> Overall, although performance does degrade as the quality of 2D masks decreases, our method remains sufficiently robust, making these negative effects acceptable.
>
> ## Reference
> [1] Grounded SAM: Assembling Open-World Models for Diverse Visual Tasks (https://github.com/IDEA-Research/Grounded-Segment-Anything)

---

> > ### Comment · Reviewer_G6nS · 2025-11-26
> >
> > I thank the authors for their detailed response. The additional clarifications and experiments involving Grounded SAM and noisy masks are very informative. As a result, I will maintain my positive rating.

---

> > > ### Author Response · Authors · 2025-11-26
> > > **Acknowledgment to Reviewer G6nS**
> > >
> > > We sincerely thank the reviewer for the constructive feedback and the positive evaluation. We are glad that the additional clarifications and experiments were helpful in addressing the concerns, and we truly appreciate your time and effort in reviewing our work.

---

### Official Review · Reviewer_LW1e · 2025-10-31

**Soundness:** 3
**Presentation:** 3
**Contribution:** 2
**Rating:** 4
**Confidence:** 3

**Summary:**

The paper proposes OVSeg3R, a training scheme that turns a closed-vocabulary 3D instance segmentor (SegDINO3D) into an open-vocabulary model by: (i) lifting per-view 2D open-vocabulary masks into 3D via reconstruction correspondences; (ii) a View-wise Instance Partition (VIP) procedure that partitions scene-level predictions to views to avoid false positives from partial labels; and (iii) a 2D Instance Boundary-aware Superpoint (IBSp) graph that prunes KNN edges crossing 2D instance boundaries before superpoint clustering. OVSeg3R reports +2.3 mAP overall on ScanNet200 (val) and +7.7 mAP on novel classes in the “standard” setting, with improved tail-class performance.

**Strengths:**

Clear training recipe that leverages modern 3D reconstruction and strong 2D open-vocabulary segmentors to create supervision without manual 3D labels; aligns with realistic inputs from videos/handheld sensors.

IBSp is a practical fix to over-smoothed reconstructions that would otherwise merge thin/low-salience objects into planes; the pruning logic is transparent

**Weaknesses:**

The pipeline inherits any bias/error from the 2D segmentor (DINO-X) and reconstruction correspondences. There is no quantitative audit of mask noise, cross-view inconsistencies, or the error rate of the VIP view assignment/truncation (only a qualitative claim that it’s “highly reliable”).

The main ablation shows VIP matters more than IBSp on the evaluation set, but the paper itself argues IBSp is crucial when reconstructions are noisy; rigorous stress tests with controlled boundary noise are missing.

The paper concedes there’s no mathematical guarantee the partition always assigns to the “right” view; no statistics on misassignments or training-time instability (e.g., oscillations/false negatives) are provided.

**Questions:**

Cost profile: Report FLOPs/params and p50/p90 latencies for SegDINO3D-VL forward, 2D sampling, IBSp graph build, and VIP,

Failure cases: Where do VIP or IBSp harm performance (thin structures, reflective surfaces, severe drift)? Show qualitative counter-examples.

Inject controlled 2D mask noise (boundary dilation/erosion, label flips) and 3D correspondence noise (perturbed C) to quantify VIP/IBSp robustness. Report mAP deltas.

---

> ### Author Response · Authors · 2025-11-23
> **Rebuttal**
>
> ## Quantitative audit of noise from 2D masks.
> As suggested by the reviewer, we inject controlled 2D mask noise with different kernel sizes for random dilation and erosion, and use Grounded SAM [1], as a replacement for DINO-X to provide 2D masks with noisy class labels. Limited by the time and resources, we conduct the experience with only about half of the iterations. The results are shown below:
>
> | Noise | mAP |
> | ----------------- | --- |
> | No Noise                      | 34.6 |
> | 0~10 pixels Noise Kernal Size | 34.6 |
> | 0~20 pixels Noise Kernal Size | 33.9 |
> | 0~40 pixels Noise Kernal Size | 33.4 |
> | Grounded SAM Masks            | 32.8 |
>
> We observe that when a small amount of noise is added to the mask boundaries (0–10 pixels), the performance of the model remains nearly unaffected. As the noise level increases further, the performance does degrade, but the drop remains within an acceptable range (-1.2 mAP).
>
> When we replace DINO-X with Grounded SAM as the 2D mask provider, the mask quality deteriorates for a different reason. Grounded SAM generates masks from the 2D bounding boxes predicted by GroundingDINO, and the produced masks do not always correspond to the exact objects intended by the boxes (one 2D box may contain multiple objects, SAM does not know which specific object within the box to segment). This misalignment results in incorrect class labels for some masks, which directly harms the open-vocabulary classification ability of the model. As shown in our results, this causes a performance drop of approximately -1.8 mAP.
>
> Overall, although performance does degrade as the quality of 2D masks decreases, our method remains sufficiently robust, making these negative effects acceptable.
>
> ## 3D correspondence noise.
>
> Although the reconstruction results from modern 3D reconstruction systems (e.g., VGGT and MASt3R-SLAM) may contain errors, the 2D pixel to 3D point correspondences provided by these algorithms are essentially error-free. This stems from the way these methods are implemented: they no longer require estimating pixel correspondences across multiple views and performing triangulation to obtain 3D points. Instead, leveraging large-scale training data, each pixel is directly learned to predict its corresponding coordinates in the world frame (typically the camera coordinate system of the first frame). As a result, there are no errors in the 2D pixel to 3D point correspondences, even if the reconstructed geometry itself is imperfect.
>
> ## Error rate of VIP during training and its effect on training-time instability.
>
> Indeed, while VIP helps reduce false positives, it can also introduce occasional false negatives. However, this typically occurs when the positive prediction itself has a low-quality mask, causing its subset in the current sub-scene to have a higher matching cost with the ground-truth subset than some negative predictions. Such cases are rare and mostly occur in the early stages of training. As training progresses, the likelihood of these occurrences decreases rapidly.
>
> Moreover, during early training, these sporadic noises are negligible compared to the overall instability at initialization. Consequently, these occasional errors do not destabilize training.
> On the contrary, by removing a large number of false positives, VIP enables OVSeg3R to train stably on pseudo-labels, with the training loss consistently decreasing. The significant performance improvement provided by VIP in Table 3 verifies this.
>
> __In addition__, we would like to emphasize that the additional meaningful supervisions provided by OVSeg3R are pseudo labels.
> As an inherent property of pseudo labels, they should not be expected to provide formal mathematical guarantees or to be completely free of noise, just like other pseudo-labeling methods.
>
> ## The importance of IBSp during inference in real-world applications.
>
> First, it is important to clarify that IBSp _does not address noise in the reconstructed point cloud; rather, it is designed to mitigate the issue of over-smoothed reconstructions_.
>
> The reason why its impact does not appear significant in the main ablation is that the point clouds provided by the evaluation set are high-quality, corrected manually, where over-smoothing is minimal. Therefore, the benefits of IBSp in handling over-smoothed regions are not reflected in these results.
>
> However, in real-world scenarios, where reconstruction algorithms often produce such over-smoothed outputs, making IBSp an important module, as evidenced in Fig. 7 in the appendix.
>
> ## Reference
> [1] Grounded SAM: Assembling Open-World Models for Diverse Visual Tasks (https://github.com/IDEA-Research/Grounded-Segment-Anything)

---

> > ### Comment · Reviewer_LW1e · 2025-11-28
> >
> > Thanks for the detailed rebuttal, and addressed most of my concens.

---

### Official Review · Reviewer_Q4jS · 2025-10-31

**Soundness:** 3
**Presentation:** 3
**Contribution:** 2
**Rating:** 6
**Confidence:** 4

**Summary:**

The paper introduces OVSeg3R, an approach to learn *open-vocabulary 3D instance segmentation* from 2D foundation models. To achieve this, it proposes an augmented version of SegDINO3D, namely SegDINO3D-VL, which replaces the categorical classifier with a CLIP-based feature predictor. To improve training stability, two additional components are introduced: (a) the View-wise Instance Partition (VIP) algorithm, which assigns each 3D mask prediction to the corresponding camera view to ensure consistent multi-view supervision, and (b) the 2D Instance Boundary-aware SuperPoint (IBSp) module, which modifies the traditional superpoint aggregation to account for 2D mask boundaries and prevent over-smoothing of 3D instance masks. In addition, the model is trained on point maps predicted either by Mast3r-SLAM or VGGT, and this makes it robust to noisy point clouds. Experiments on ScanNetV2 and ScanNet200 demonstrate that OVSeg3R achieves state-of-the-art performance, outperforming existing open-vocabulary approaches and even several fully supervised baselines.

**Strengths:**

1. **SoTA performances on Open-Vocabulary Instance Segmentation.** By carefully extending SegDINO3D to the open vocabulary OVSeg3R achieves SoTA performances for the task. This makes the method valuable for the research community working on multimodal 3D understanding.
2. **Evaluation of the design choices.** The paper includes ablation studies that analyze the impact of the VIP algorithm and the IBSp module, providing useful insights into their contribution to the overall method.
3. **Robustness.** Training on point clouds produced by VGGT and Mast3rSLAM makes the method robust to reconstruction noise. This property is practically important, as it allows faster deployment without relying on possibly slow, high-accuracy 3D reconstruction pipelines.

**Weaknesses:**

1. **Limited methodological novelty.** While the task is interesting and the results surpass existing baselines, I am not entirely convinced about the technical novelty of the method. The core idea of distilling 2D vision-language knowledge into 3D has already been explored in prior works such as OpenScene, OpenMask3D, etc. In addition, the two extensions proposed in the paper (i.e. VIP  algorithm and IBSp module) can be seen more as implementation choices than technical contributions.

**Questions:**

1. In line 234 what does $s$ represents?
2. How well does the model perform when given higher resolution reconstructions?

---

> ### Author Response · Authors · 2025-11-23
> **Rebuttal**
>
> ## Comparison with methods distilling vision-language knowledge from 2D.
>
> Indeed, as summarized in the introduction section, there have been some methods exploring how to enhance 3D perception models with the well-studied 2D models [1,2,3]. However, although the accurate correspondences between 2D pixels and 3D points can be easily obtained, establishing reliable correspondences between 2D instances and 3D instances is significantly more challenging. This makes prior methods stop at distilling open-vocabulary semantic classification ability from 2D models (semantic segmentation task), leaving their native 3D instance-level perception ability (instance segmentation task) underdeveloped. More specifically:
>
> 1. OpenMask3D projects class-agnostic 3D masks into 2D to obtain 2D features, and then retrieves categories from the 2D branch. Thus, the 3D backbone does not actually internalize the open-vocabulary capability; it merely queries 2D features as an external knowledge source. As a consequence, while it can classify novel objects, it still struggles to detect them, leading to systematic missing detections when the object category is unseen during training.
> 2. OpenScene and subsequent semantic distillation methods focus solely on point-wise semantic labeling. These methods distill 2D vision-language knowledge only at the classification level, without learning 3D instance boundaries. Therefore, they do not address the more challenging problem of open-vocabulary 3D instance segmentation, where correct classification and accurate instance mask prediction are both required.
>
>
> ## Technical contribution of extensions under our main idea.
>
> For the Instance Boundary-aware Superpoint (IBSp), it is motivated by our observation of the over-smoothed reconstruction results and is decoupled from OVSeg3R. IBSp can benefit any segmentation approach that want to operate on reconstructed scenes, regardless of the architecture or supervision.
>
> For the View-wise Instance Partition (VIP), we have to acknowledge that it is indeed coupled with our proposed training scheme. However, it is not an optional heuristic, but an indispensable component that enables stable learning from 2D pseudo-labels.
> VIP plays a role similar to the masking strategy in MAE and the KL term in VAE reconstruction, which are not the central conceptual novelties of their frameworks, but are indispensable stabilizers that prevent collapse and make the learning signal truly effective.
>
> ## Performance on higher resolution reconstructions.
>
> From the segmentation results we have examined, the model tends to perform better when the reconstructed point clouds are of higher resolution, and finer geometry leads to more accurate instance boundaries. However, due to the absence of an annotated benchmark for visual-SLAM reconstructions, we are unable to report numerical metrics to support this observation at present.
>
> ## Notation declaration.
>
> _s_ denotes the list of class names predicted by the 2D segmentor for each object mask. We will provide a more precise and explicit definition of s in the camera-ready version for clarity.
>
> [1] OpenScene: 3D Scene Understanding with Open Vocabularies
>
> [2] PLA: Language-Driven Open-Vocabulary 3D Scene Understanding
>
> [3] OpenMask3D: Open-Vocabulary 3D Instance Segmentation

---

### Author Response · Authors · 2025-11-21
**Response Statement and Acknowledgment to Reviewers**

We are sincerely grateful to the reviewers for dedicating their time and effort to thoroughly evaluate our work. The constructive suggestions and thoughtful concerns raised in the reviews are highly valuable and greatly help us improve the paper.

However, due to a tight project schedule on the company side during the rebuttal period, we are unfortunately unable to provide the full responses at this time. We will provide our responses to each reviewer’s questions by the end of this weekend (November 23, 2025). We would like to express our sincere gratitude for the reviewers’ understanding and patience.

---

### Author Response · Authors · 2025-11-23
**Rebuttal**

We sincerely thank the reviewers once again for their valuable time and constructive feedback. Due to the tight project schedule during the rebuttal period, we are unable to conduct all additional experiments requested. Nevertheless, we have carefully prepared responses to all of the main questions and hope our explanations clarify the contributions and significance of our work.

To assist the reviewers and the Area Chair, here we summarize the main recognition of our work provided by each reviewer:

*  OVSeg3R addresses a critical problem, enabling methods robust to reconstruction noise. Important for real-world deployment and application __(Q4jS, G6nS, LW1e)__. Indicating a real progress towards robust real-world 3D perception __(3Khs)__.
*  The motivation is clear __(G6nS)__, and the proposed OVSeg3R provides an inovative supervision strategy __(3Khs)__.
*  Extending SegDINO3D to open-vocabulary and surpassing previous SoTA methods by a large margin, valuable for multimodal 3D understanding. __(Q4jS, LW1e, G6nS, 3Khs)__
*  Correctly identify two main issues, and propose respective solutions __(G6nS)__, their corresponding ablation provides useful insights __(Q4jS)__,

| Reviewer   | Q4js | G6nS (Before & After Rebuttal) | 3KhS (After Rebuttal) | LW1e |
| --------   | ---- | ---- | ---- | ---- |
| Rating     | 6    | 6    | 6    | 4    |
|Confidence  | 4    | 3    | 3  | 3    |

---

### Author Response · Authors · 2025-11-30
**Summary of Rebuttal Progress Before the Information Leak Incident**

Dear AC and Reviewers,

We are sorry to learn about the recent information leak on OpenReview and the resulting impact on the reviewing process of ICLR. We understand that this unexpected incident has introduced additional workload, and we would like to express our sincere appreciation to the AC for the extra effort required during this challenging situation.

To help reduce the burden on the AC, we summarize below the key progress that was explicitly acknowledged by the reviewers during the two-week rebuttal period prior to the information leak incident:

1. Addressing Reviewer 3KhS’s concerns

	•	The generalization ability of OVSeg3R across different 3D instance segmentation backbones.

	•	Whether the performance gain originates merely from introducing more pseudo labels, or from the proposed modules that make pseudo labels truly effective.

	•	The need for additional metrics to make the open-vocabulary claim even more convincing.


    Reviewer's response to rebuttal (raise rating from 4 to 6):
    > “Thank you for the rebuttal. After reading the other reviews and the authors’ responses, I believe my concerns have been addressed, and I will raise my rating to 6.”

2. Addressing Reviewer G6nS’s concerns

	•	The domain discrepancy between training and evaluation.

	•	Potential failure modes introduced by the proposed key module (VIP).

	•	The generalization capability of the 2D perception model involved in our pipeline.

    Reviewer's response to rebuttal (maintain the positive rating 6):
    > “I thank the authors for their detailed response. The additional clarifications and experiments involving Grounded SAM and noisy masks are very informative. As a result, I will maintain my positive rating.”

---

### Meta-Review · Area_Chair_wTo1 · 2026-01-07

**Summary:**

This paper receives 2x marginal accept and 2x marginal reject at the initial reviews. The strengths of the paper are: 1) SoTA performances on Open-Vocabulary Instance Segmentation. 2) Addresses a critical problem with strong motivation. 3) Innovative Supervision Strategy. 4) Targeted solution; 5) Strong experimental results for evaluation of the design. Significant gains on long-tail and novel categories. Benchmark thoroughness. 6) Robustness. The weaknesses are: 1) pipeline inherits any bias/error from the 2D segmentor (DINO-X) and reconstruction correspondences. 2) no mathematical guarantee the partition always assigns to the “right” view; no statistics on misassignments or training-time instability; 3) lacks clarity in technical details. 4) Reliance on a Single 2D Perception Model. 5) Limited Generalizability Testing. These issues are however clarified during the rebuttal and discussion phases. One of the reviewers who gave a marginal reject commented "Thanks for the detailed rebuttal, and addressed most of my concerns". The other reviewer who gave a marginal reject mentioned "I believe my concerns have been addressed, and I will raise my rating to 6". Considering the strengths and weaknesses, the rebuttal and the discussions, the AC follows the suggestions of the reviewers to accept the paper.

**Reviewer Concerns:**

The weaknesses are: 1) pipeline inherits any bias/error from the 2D segmentor (DINO-X) and reconstruction correspondences. 2) no mathematical guarantee the partition always assigns to the “right” view; no statistics on misassignments or training-time instability; 3) lacks clarity in technical details. 4) Reliance on a Single 2D Perception Model. 5) Limited Generalizability Testing. These issues are however clarified during the rebuttal and discussion phases. One of the reviewers who gave a marginal reject commented "Thanks for the detailed rebuttal, and addressed most of my concerns". The other reviewer who gave a marginal reject mentioned "I believe my concerns have been addressed, and I will raise my rating to 6".

**Reviewer Scores:**

I believe that the two marginal accepts will remain and the two marginal rejects will be raised to at least marginal accepts. There are also no further majors raised during the discussion phase.

---

### Decision · Program_Chairs · 2026-01-26

Accept (Poster)